# When the Beetles Hit the Fan: The *Fan-Trap*, an Inexpensive, Light and Scalable Insect Trap under a *Creative Commons* License, for Monitoring and Experimental Use

**DOI:** 10.3390/insects13121122

**Published:** 2022-12-05

**Authors:** Jean-Claude Grégoire, Emilio Caiti, Séverine Hasbroucq, Jean-Marc Molenberg, Sylvain Willenz

**Affiliations:** 1Spatial Ecology Lab. (SpELL), CP 160/12, Université Libre de Bruxelles, 50 Av. F.D. Roosevelt, 1050 Bruxelles, Belgium; 2Evolutionary Biology & Ecology Unit (EBE), Université Libre de Bruxelles, CP 160/12, 50 Av. F.D. Roosevelt, 1050 Bruxelles, Belgium; 3Agroecology Lab., CP 264/2, Université Libre de Bruxelles, Blvd. du Triomphe, 1050 Bruxelles, Belgium; 4Sylvain Willenz Design Office, 99, Vieille Rue du Moulin 1180, 1180 Uccle, Belgium

**Keywords:** traps, beetles, monitoring, surveys, spatial distribution

## Abstract

**Simple Summary:**

There is a need for cheap, easily deployed traps for insect pest monitoring. Here, we propose the design of an inexpensive *fan-trap*, under a Creative Commons BY-SA License. Using the blueprint we provide, anyone could laser cut their own traps from a sheet of polypropylene or have this done commercially by a contractor. As they are flat when unfolded, the fan-traps ship easily. When mounted, they are easy to transport in the field in a backpack. The blueprint can also be modified in order to resize the traps to adapt them for different purposes.

**Abstract:**

Monitoring is an important component in pest management, to prevent or mitigate outbreaks of native pests and to check for quarantine organisms. Surveys often rely on trapping, especially when the target species respond to semiochemicals. Many traps are available for this purpose, but they are bulky in most cases, which raises transportation and deployment issues, and they are expensive, which limits the size and accuracy of any network. To overcome these difficulties, entomologists have used recycled material, such as modified plastic bottles, producing cheap and reliable traps but at the cost of recurrent handywork, not necessarily possible for all end-users (e.g., for national plant-protection organizations). These *bottle-traps* have allowed very large surveys to be conducted, which would have been impossible with standard commercial traps, and we illustrate this approach with a few examples. Here, we present, under a *Creative Commons* BY-SA License, the blueprint for a *fan-trap*, a foldable model, laser cut from a sheet of polypropylene, which can rapidly be produced in large numbers in a Fab lab or by a commercial company and could be transported and deployed in the field with very little effort. Our first field comparisons show that *fan-traps* are as efficient as *bottle-traps* for some Scolytinae species and we describe two cases where they are being used for monitoring.

## 1. Introduction

Insect surveys are necessary at different spatial scales to determine the extent and magnitude of native pest populations, forecast outbreaks, delimit critical areas and assess the success of control operations. Surveys also allow for population assessments and spread measurements in biological and ecological research. Non-native, invasive pests expand their distribution range on their own via flight or moving with travelers or with commercial goods (e.g., unprocessed roundwood or live plants for planting), with wood packaging material or as hitchhikers in vehicles or containers. This also creates a need for cheap and easy-to-handle monitoring tools.

In many countries, the inspection services of national plant-protection organizations (NPPOs) monitor high-risk locations (ports, airports, road, railway hubs, commercial nurseries, companies importing stones or machinery in wood packaging material) [1]. Surveys can be operated visually, but the large amount of incoming goods makes this very difficult and unreliable. Recently, in March 2021, the public witnessed an example of the magnitude of the problem when the 400 m long giant container ship *Ever Given* blocked Egypt’s Suez Canal for almost one week. This ship carries 20,400 6 m long cargo containers. Yet, even larger ships are active that can carry up to 24,000 containers. Although all containers do not contain material at risk, the scale of commercial shipment activity makes visual inspections extremely inefficient. In addition to visual inspections, when chemical or visual attractants are available, inspectors can also deploy traps in entry points and in their vicinity ([2,3,4,5,6,7,8]). This raises the technical and commercial issues of traps—they should be inexpensive, easy to handle and efficient. Here, we present an inexpensive trap model for surveying beetles (Coleoptera), which constitute one of the major groups of invasive insect pests and are easily transported in woody materials (crates, dunnage, pallets) used in trade. These traps have been tested so far on bark and ambrosia beetles (Curculionidae, Scolytinae) and on Buprestidae.

### 1.1. Present State of the Art: Available Commercial Traps

A variety of commercial traps is currently on the market and used in the surveys described above. They include *Theysohn* trap, *Witasek* traps, *Ecotraps*; *Lindgren funnel traps*; *Crosstraps* and *Crosstraps Mini*. They are, in general, very efficient in catching large numbers of beetles (see, e.g., [2,3,4,5,6,7,8,9,10]) but they weigh several kilograms and are bulky. They cannot be easily transported in large numbers (one person can walk with one or two of them at a time, which makes it difficult to establish large trap networks offroad, and they take time and manpower for set up (often, two persons are necessary)). Further, in many cases, these traps must be fixed on stakes or hung on ropes attached to two adjacent trees. Their individual prices range from EUR 25 to 50 (information generally available on the producers’ websites).

### 1.2. A Cheap and Versatile Alternative: Bottle-Traps

There are many examples in the literature of makeshift traps made from plastic bottles and used in experimental work. Rieske and Raffa [11] used baited flight traps made from modified gallon-sized milk plastic containers, positioned upside down with three sides removed for monitoring *Pissodes nemorensis* and *P. strobi* (Coleoptera, Curculionidae). They found that these traps were more effective than baited pitfall traps. Steiniger et al. [12] showed that traps improvised from 2 L plastic water bottles, equipped with commercial alcoholic disinfectants, are very efficient and cheap for monitoring invasive ambrosia beetles (Coleoptera, Curculionidae, Scolytinae). Reding et al. [13] used ethanol-baited bottle traps to monitor *Xylosandrus crassiusculus* and *Xylosandrus germanus*. Olenici et al. [14] monitored *X. germanus* in Romania with similar traps.

Bottle traps have been used since the early 2000s in our own research in Belgium, France and Britain. These traps each consist of a 2 L commercial transparent PET bottle, cut longitudinally and turned upside down to form a 21 × 13 cm interception pane, the inverted bottleneck serving as a collecting funnel connected to a 50 mL clear polystyrene collecting tube, which can be filled with propylene glycol as a preservative. Franklin et al. [15] and Franklin and Grégoire [16] used these traps for release–recapture experiments with *Ips typographus* (Coleoptera, Curculionidae, Scolytinae) in Belgium; Grégoire et al. [17] monitored the presence and abundance of various potentially harmful Scolytinae in the Forêt de Soignes, near Brussels, with 100 traps spread across 1600 ha; Meurisse et al. [18] used bottle traps for monitoring and release–recaptures of *Rhizophagus grandis* (Coleoptera, Monotomidae) in Belgium and France; Piel et al. [19,20] monitored *I. typographus* in the city of Brussels and *Ips duplicatus* in Liège; Warzée et al. [21] surveyed *Thanasimus formicarius* (Coleoptera, Cleridae) in the Vosges, France. The Observatoire Wallon de la Santé des Forêts has maintained, since 2012, a permanent network of 50 bottle traps across Wallonia for monitoring *I. typographus* [22]. A review of interception traps (including bottle traps) was provided by the late Simon Leather [23].

Other studies (Grégoire et al., unpublished to date) involved networks of 460 bottle-traps in Wallonia (2005–2006) and of 300 bottle-traps in a transect from Wallonia to the Champagne area in France. Bottle-traps were also used in large numbers (20–45 traps/tree) for passive interception to monitor random landing of bark beetles on already attacked and healthy spruce trees in a stand colonized by *Dendroctonus micans* in Lozère (France). Inward et al. (in preparation) used an extensive network of 185 and 250 bottle-traps in 2021 and 2022, respectively, in France and England to measure the expansion into England of the French and Belgian *I. typographus* populations.

These cheap and efficient traps are much easier to transport and use than the commercial models available so far. At least thirty traps can be transported together in a backpack, and each trap can be fixed to a range of standing objects (trees, electricity posts, public lighting) with staples or Colson nylon binders. One problem, though, is that these three-dimensional traps are still bulky in large shipments. However, the major difficulty is that empty plastic bottles must be found or bought and the traps must be handmade, which is not practical for many potential end users, such as NPPOs. To address this resource gap, our team designed a novel model for a foldable *fan-trap* as an inexpensive, compact and open-source device option for insect monitoring. These *fan-traps* have been tested so far on Scolytinae and on Buprestidae.

## 2. The *Fan-Trap*—A Description

The *fan-traps* are described under a Creative Common BY-SA licence (CC BY-SA: This license allows reusers to distribute, remix, adapt and build upon the material in any medium or format, so long as attribution is given to the creator (https://creativecommons.org/licenses/by-sa/4.0/ accessed on 17 October 2022)). The full description and blueprint files are available at the following sites: https://ebe.ulb.ac.be/ebe/Fan-trap.html accessed on 17 October 2022; https://spell.ulb.be/fan-trap_description.zip accessed on 17 October 2022. A detailed description of the traps, including a detailed blueprint and practical details of trap manufacturing with a laser cutter, can also be found in the File S1 of the Appendix A.

A *fan-trap* (Figure 1 and Figure 2) consists of an arrow-shaped polypropylene sheet that can be folded into a funnel by fitting lugs on one side of the “arrowhead” into corresponding slots on the other side. The bottom of the funnel is inserted into a 2.5 cm diameter circular hole cut into the screw cap of a collecting container. The container can either be kept dry or partly filled with a preservative liquid (e.g., propylene glycol). The cap is physically attached to the trap with pre-cut wedges, bent outwards at the bottom of the funnel. When servicing the traps, the containers can be unscrewed, tightly closed with unperforated caps and replaced with a new collection container. Different sizes of collection containers can be used depending on the expected catch yields. When folded, the traps can be attached to trees or any other support (e.g., electrical posts) with plastic ties or strings pulled through pre-cut fixation holes or with staples. Additional fixation holes can be added to attach the lures (see Appendix A). For added performances, the traps could be sprayed with polytetrafluoroethylene (PTFE).

The traps used in our experiments were cut out of a 0.8 mm thick polypropylene sheet using a Metaquip MQ1590 laser cutter driven by the RDWorks 8.0 software (https://rdworks.software.informer.com/8.0/ (accessed on 25 August 2022)). Previous tests showed that thicker sheets (maximum 1 mm) could be used as well. Polypropylene was chosen due to its affordable price, flexibility and structural durability, allowing one to fold and unfold the traps for multiple successive uses. The traps are machine-washable and tolerate high temperatures up to 135–165 °C, depending on the polymer [24]. The traps can be cut in a Fab lab or procured from a commercial company provided with the blueprint. Considering all expenses (polypropylene sheets, manpower and, in our case, Fab lab fees, each of the *fan-traps* we produced so far cost less than EUR 2). Further, 75 *fan-traps* could be processed within one hour (for details, see Appendix A).

## 3. Experimental Support

### 3.1. Material and Methods

#### 3.1.1. Comparisons with Bottle-Traps

An experiment was set up in the Vosges mountains, département du Bas-Rhin, massif du Donon (48.4403, 7.1091) in 2005, at a time when the area was still a hotspot for *I. typographus* activity after the Lothar storm in December 1999. Fifteen blocks were spread out in a meadow bordering spruce stands. Each block comprised one fan-trap (size of the trapping panel: width = 15 cm; height = 20 cm) and one bottle-trap (17 cm × 19 cm), stapled to a broadleaf tree or a fir, facing East and distant from each other by ca. 5 m. The blocks were distant from each other by ca. 15 m. Each trap was baited with a Pheroprax lure (commercial lure for *I. typographus*, produced by BASF), replaced monthly. The traps were set up on 5 May 2005, visited on 1 June, 6 July, 10 August and 16 September.

#### 3.1.2. Comparisons with Bottle Traps and Commercial Trap Models

This experiment was run in 2005 in Wellin, province of Luxembourg, Belgium (50.0581, 5.1214) in a clear-cut area adjacent to mature spruce stands with sporadic attacks. The traps compared were *fan-traps*, *bottle traps*, *Theysohn* (W × L: 60 cm × 50 cm) and *Intercept* traps (four panels, each 15 cm × 70 cm). The objective was to compare trap catches of different species of Scolytinae, within an optimal period for each of them, with no regard to monitoring populations. In a first run, we aimed at trapping xylomycetophagous Scolytinae (*Trypodendron* spp. and *Anisandrus dispar*), so all trap types were baited, from April 4 to May 12, 2005, with lineatin (Witasek, Feldkirchen in Kärnten, Austria) and ethanol (97.1% ethanol, 2.9% ether; diffusion rate ca 1.2 g/d) and were serviced on May 12. In a second run aimed at trapping *I. typographus*, the traps were baited from 12 May to 10 June 2005, with Pheroprax, and were serviced on 27 May and 10 June. The lures were renewed each time the traps were serviced. The peak flight of the xylomycetophagous Scolytinae occurred during the first run, and the peak of the first flight of *I. typographus* occurred during the second run (see Appendix A). The traps were positioned along two parallel lines with 20 m distance between and each comprising twenty traps (five replicates of the four trap models, fixed on wooden stakes) placed randomly 20 m from each other. The order of the traps within each line was modified randomly at each visit.

### 3.2. Results

#### Comparisons with Bottle-Traps

The detailed results are available as Appendix A and summaries are provided in Table 1 and Table 2.

After a log(n + 1) transformation needed to stabilize the variances, a one-way analysis of variance was applied on the total catches per species for each trap model. There were no significant differences between traps for *A. dispar* (F_3,36_ = 1.849; *p* = 0.156). For all the other species, the four traps performed differently: *I. typographus* (F_3,36_ = 80.787; *p* < 0.001); *T. domesticum* (F_3,36_ = 19.931; *p* < 0.001); *T. signatum* (F_3,36_ = 10.246; *p* < 0.001); *T. lineatum* (F_3,36_ = 13.016; *p* < 0.001.

For *I. typographus* and *T. domesticum*, Student–Neuman–Keuls post hoc tests (α = 0.05) separated two homogeneous categories: the *bottle-traps* and *fan-traps* on the one hand and the *Intercept* and *Theysohn* traps on the other hand. Roughly, the two latter traps caught on average ten-times more *I. typographus* than the first two trap models. The differences were not as wide for *T. domesticum.* For *T. signatum*, there was no significant difference between the *bottle*- and *fan-traps*, and between the *fan-* and *Theysohn* traps. The *Intercept* traps performed significantly better than the other three models. For *T. lineatum*, there were no significant differences between the *bottle*-, *fan-* and *Theysohn* traps and, again, the *Intercept* traps performed significantly better than the other three models.

These differences are at least partly linked to the size of the traps. The *fan-traps* have a 20 × 15 cm, single-faced interception panel (300 cm^2^); the single-faced interception panel of the *bottle-traps* (19 × 17 cm) covers 323 cm^2^, the two-faced *Theysohn* traps (60 × 50 cm) cover 6000 cm^2^, and with their four (70 × 15 cm) two-faced panels, the *Intercept* traps, cover 8400 cm^2^. From this standpoint, the standardized trap performances (captures per cm^2^) of the *bottle*- and *fan-traps* are higher than those of the larger trap models (Table 2).

There were significant differences between traps for all species: *I. typographus* (F_3,36_ = 9.524; *p* < 0.001); *T. domesticum* (F_3,36_ = 15.459; *p* < 0.001); *T. signatum* (F_3,36_ = 64.623; *p* < 0.001); *T. lineatum* (F_3,36_ = 33.890; *p* < 0.001; *A. dispar* (F_3,36_ = 7.734; *p* = 0.156). The *bottle-* and *fan-traps* caught more insects per cm^2^ than the two commercial models. For *I. typographus*, *T. domesticum*, *T. lineatm* and *X. dispar*, Student–Neuman–Keuls post hoc tests (α = 0.05) separated two homogeneous categories: the *bottle-traps* and *fan-traps* on the one hand and the *Intercept* and *Theysohn* traps on the other hand. For *T. signatum*, there were no significant differences between the *bottle*- and *fan-traps*, and the performances of the two other models were significantly lower and significantly differed from each other.

## 4. Discussion

Individually, as seen above with *I. typographus*, but depending on the target species, the *fan-traps* might catch far fewer insects than commercially available traps. This, however, does not necessarily mean that they are less efficient collectively, considering that their price and ease for transport and handling allow many more of them to be deployed. As an example, during a survey of *I. typographus* in France in 2022, the deployment of 90 *bottle-traps* (similar to the *fan-traps* as regards handling time in the field) in 18 sites over 750 km took two days for a team of two persons (Caiti and Hasbroucq, unpublished).

An interesting outcome of the standardized trapping data (captures/cm^2^) is the stark difference between the two smaller and the two larger trap models, perhaps due to a propensity of the species involved in the experiment to concentrate around the lures’ location. This suggests that resizing traps should not exceed a certain limit. Byers et al. [25] reported a positive link between captures of *I. typographus* and the diameter of sticky cylindrical pheromone traps, up to a 30 cm radius. Because they are scalable, further research with *fan-traps* of increasing sizes could easily contribute to establishing optimal trap dimensions for any given species. The ease of modifying the original blueprint towards larger models would facilitate this approach.

At this point, it could prove fruitful to distinguish between the possible uses of trapping networks. Two different objectives can be considered: (i) monitoring for still-absent or rare species (surveys for potentially incoming non-native species); (ii) measuring population trends (e.g., phenology, spread or density changes) among established, often abundant, species (in particular, native pests).

For monitoring potentially incoming non-native species, one needs to reach a high probability (sometimes prescribed by the national regulatory agencies, e.g., [26,27,28,29]) to detect an unwanted organism above a certain level of prevalence. As discussed above, an optimal trap size should be adopted in this respect. Another limiting factor here is the attraction radius of the lure in the trap, i.e., the radius around the trap within which randomly flying beetles can detect components of the chosen lure. For many beetles, these attraction radii are comparatively short. Byers [30] calculated a radius of 1.4 to 16 m for *I. typographus*, but Franklin and Grégoire [16] provide a slightly larger estimate (at least 50 m). From a network of Multifunnel traps baited with frontalin and turpentine, Turchin and Odendaal [31] calculated that a trap attraction range for *Dendroctonus frontalis* is about 0.1 ha. Jactel et al. [32] calculated a 92–123 m radius for *Monochamus galloprovincialis*, whilst Torres-Vila et al. [33] reported a 50 m radius for the same species. These rather short distances mean that, for maximum monitoring accuracy, traps should be deployed at densities high enough to avoid gaps between their respective areas of attractiveness. For example, *D. frontalis* should be monitored with traps positioned 20 m from each other. This would be in favor of deploying many cheaper but perhaps less efficient traps.For measuring population trends in well-established species, the trapping results are compared to each other, e.g., against time to measure phenology or density changes, or against space to measure population expansion. The trapping data in this case are relative (one trap against the others) and even small catches could be compared to each other.

As some target species have distinct color preferences [34,35], the trap could be painted or procured by commercial companies that could provide and cut polypropylene material in any color. Trap positioning could also depend on the targets’ preferences. In an ongoing project aiming at monitoring Buprestidae, pairs of *fan-traps* painted green or yellow, sprayed with PTFE and attached together back-to-back, were suspended in the crown of broadleaf trees at a height of 5 to 15 m with nylon ropes. Strings of two or more of these paired traps could be attached to the same rope. Thus, 128 paired traps were deployed in 2022 (180 pairs in Belgium, 24 in France, 12 in Canada and 12 in the US). Their deployment was not more difficult than that of commercial funnel-traps or prism-traps, and they resisted similarly, or even better, against wind and rain.

The *fan-traps* were successfully tested so far only with small- (<1 cm) or medium-sized (1–2 cm) Coleoptera: respectively, Scolytinae and Buprestidae. The capacity of *fan-traps* to catch larger beetles must still be tested. Other taxa, e.g., Lepidoptera or Diptera, are not very likely targets, because they could probably veer off, or land, when hitting the vertical panel. The proportionally large wings of the moths and butterflies might also prevent them from falling into the collecting pots.

## Figures and Tables

**Figure 1 insects-13-01122-f001:**
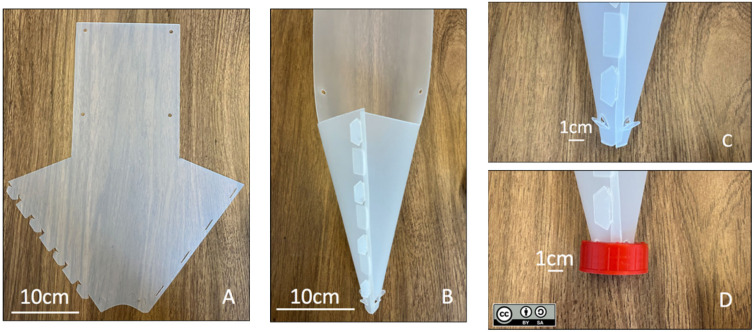
An unfolded and an assembled *fan-trap*. Details of the collecting-container cap attachment. (**A**)—The flat design; (**B**)—folded trap; (**C**)—pre-cut wedges used for attaching the collecting container’s cap; (**D**)—cap attached to the bottom of the trap.

**Figure 2 insects-13-01122-f002:**
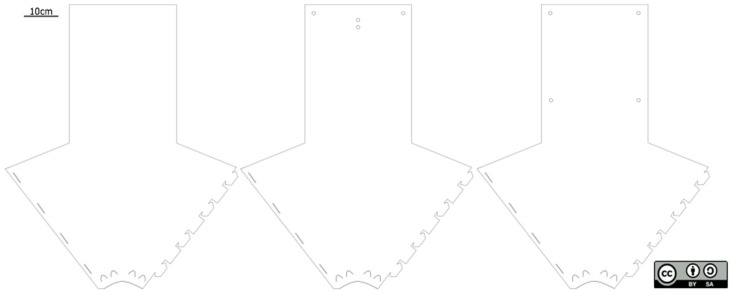
Blueprint of the *fan-trap* with various attachment hole patterns.

**Table 1 insects-13-01122-t001:** Total catches per trap across the whole trapping period for each of the two groups of Scolytinae (the “xylomycetophagous” species, i.e., *Trypodendron lineatum*, *T. domesticum*, *T. signatum* and *Ips typographus*). For each species, the traps are listed in decreasing order according to trapping results.

Species	Trap	N	Mean	SEM
*Ips typographus*	Intercept	10	2106.70	327.35
	Theysohn	10	1965.70	276.46
	Bottle-traps	10	252.30	34.08
	Fan-traps	10	229.80	40.77
*Trypodendron domesticum*	Intercept	10	30.60	5.33
	Theysohn	10	22.70	4.38
	Bottle-traps	10	7.10	1.21
	Fan-traps	10	6.40	1.07
*T. signatum*	Intercept	10	292.90	72.88
	Theysohn	10	166.90	18.51
	Fan-traps	10	115.20	10.83
	Bottle-traps	10	100.50	8.75
*T. lineatum*	Intercept	10	111.70	28.09
	Theysohn	10	50.10	4.39
	Fan-traps	10	35.20	3.97
	Bottle-traps	10	32.30	4.55
*Anisandrus dispar*	Theysohn	10	77.20	19.64
	Fan-traps	10	59.80	16.60
	Intercept	10	45.00	7.54
	Bottle-traps	10	39.50	12.10

**Table 2 insects-13-01122-t002:** Standardized total catches per trap (catches/cm^2^) across the whole trapping period for each of the two groups of Scolytinae (*Ips typographus*, and the “xylomycetophagous” species, i.e., *Trypodendron lineatum*, *T. domesticum*, *T. signatum* and *X. dispar*). For each species, the traps are listed in decreasing order according to trapping results.

Species	Trap	N	Mean	SEM
*Ips typographus*/cm^2^	Bottle-traps	10	0.781	0.106
	Fan-traps	10	0.766	0.136
	Theysohn	10	0.328	0.046
	Intercept	10	0.251	0.039
*Trypodendron domesticum*/cm^2^	Bottle-traps	10	0.022	0.004
	Fan-traps	10	0.021	0.004
	Theysohn	10	0.004	0.001
	Intercept	10	0.004	0.001
*T. signatum*/cm^2^	Fan-traps	10	0.384	0.036
	Bottle-traps	10	0.311	0.027
	Intercept	10	0.035	0.009
	Theysohn	10	0.028	0.003
*T. lineatum*/cm^2^	Fan-traps	10	0.117	0.013
	Bottle-traps	10	0.100	0.014
	Intercept	10	0.013	0.003
	Theysohn	10	0.008	0.001
*Anisandrus dispar*/cm^2^	Bottle-traps	10	0.122	0.037
	Fan-traps	10	0.199	0.055
	Theysohn	10	0.013	0.003
	Intercept	10	0.005	0.001

## Data Availability

All data are available in the Appendix A.

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
