# Peer review of "When the Beetles Hit the Fan: The Fan-Trap, an Inexpensive, Light and Scalable Insect Trap under a Creative Commons License, for Monitoring and Experimental Use"

_insects, 2022, doi:10.3390/insects13121122_

Round 1

Reviewer 1 Report

Generally well written. Please revise to avoid pronoun usage, which obscures your meaning/intent.  The manuscript results would benefit from additional analyses (results could be included within existing table) that illustrates trap capture results on the basis of surface area by time for the different trap styles tested. Doing so would assist the reader in decision making between unit cost, transportability, and different trap style's expected yield.

Author Response

Dear Reviewer 1,

Many thanks for your detailed and thoughtful comments and helpful suggestions. You will find how we responded to them in the uploaded files "insects-2007323-review_1.pdf", and "Response to the Reviewers' comments - 27 November 2022.docx"

Thank you again, and best regards,

JC Grégoire

Author Response

Dear Reviewer 2,

Thank you for your useful comments. You will find how we responded to them in the attached file "Response to the Reviewers' comments - 27 November 2022.docx"

With best regards,

JC Grégoire

Round 2

Reviewer 2 Report

The authors followed the instructions of the reviewers and editor. I therefore recommend publication of the manuscript.